# Eriocitrin Inhibits Angiogenesis by Targeting VEGFR2-Mediated PI3K/AKT/mTOR Signaling Pathways

**DOI:** 10.3390/nu16071091

**Published:** 2024-04-08

**Authors:** Ji-Yoon Baek, Jeong-Eun Kwak, Mok-Ryeon Ahn

**Affiliations:** 1Department of Health Sciences, The Graduate School of Dong-A University, Busan 49315, Republic of Korea; byoon0403@gmail.com (J.-Y.B.); kpj781120@naver.com (J.-E.K.); 2Department of Food Science and Nutrition, College of Health Sciences, Dong-A University, Busan 49315, Republic of Korea; 3Center for Food & Bio Innovation, Dong-A University, Busan 49315, Republic of Korea

**Keywords:** eriocitrin, angiogenesis, VEGFR2, apoptosis, HUVECs, tube formation

## Abstract

Eriocitrin, a flavanone found in peppermint and citrus fruits, is known to possess many physiological activities. However, the anti-angiogenic effects of eriocitrin are yet to be fully elucidated. Therefore, the objective of this research was to explore the anti-angiogenic effects of eriocitrin both in vitro and in vivo as well as its underlying mechanism. Anti-angiogenic effects of eriocitrin were evaluated utilizing in vitro models of angiogenesis, including inhibition of tube formation, and induction of apoptosis in human umbilical vein endothelial cells (HUVECs). A chorioallantoic membrane (CAM) assay in chick embryos was also performed to evaluate the in vivo effects of eriocitrin on angiogenesis. Results showed significant eriocitrin effects on proliferation, tube formation, migration, and apoptosis in HUVECs. Furthermore, in vivo analysis revealed that eriocitrin significantly suppressed the formation of new blood vessels. In particular, it regulated MAPK/ERK signaling pathway and VEGFR2, inhibited the downstream PI3K/AKT/mTOR signaling pathway, and activated apoptosis signals such as caspase cascades. In HUVECs, the expression of matrix metalloproteinases (MMP-2 and MMP-9) exhibited an inhibitory effect on angiogenesis through the suppression of the signaling pathway. Therefore, eriocitrin presents potential for development into an antiangiogenic therapeutic agent.

## 1. Introduction

Angiogenesis is crucial for tumor growth and metastasis. It refers to the creation of new blood vessels from preexisting ones. It is important for tumor growth because tumors need blood vessels to get oxygen and nutrients. Angiogenesis plays a critical role in both tumor growth and metastasis [1]. The angiogenesis process involves a series of intricate steps, including endothelial cell proliferation, migration, infiltration, basal membrane degeneration, and the formation of new tubes from existing blood vessels [2]. Vascular endothelial growth factor (VEGF) plays a crucial role as an angiogenesis factor, exerting significant activity within endothelial cells. VEGF receptor 2 (VEGFR2) is among the most significant and functionally crucial receptors within the VEGF family. It primarily regulates the proliferation, migration, and tube formation of endothelial cells [3]. VEGFR2 activity activates downstream, including MAPKs and PI3K/AKT signaling pathways, which leads to tumor formation [1,4]. Inhibiting phosphorylation of VEGFR2 affects the development and growth delay of cancer by inhibiting angiogenesis. The primary mechanisms of angiogenesis inhibitors involve inducing apoptosis of vascular endothelial cells, which are pivotal in angiogenesis [5]. Therefore, anti-angiogenesis strategies, capable of inhibiting VEGFR2 signals and selectively inducing apoptosis solely in vascular endothelial cells, have garnered considerable attention in tumor prevention.

Eriocitrin, also known as eriodictyol 7-rutinoside, is a flavanone compound abundant in peppermint, lemon, and various citrus fruits [6,7]. Eriocitrin exhibits potent biological effects, including its strong antioxidant [8], antitumor [9,10], anti-allergic [11], and anti-inflammatory properties [12,13,14]. Recent studies have shown that eriocitrin can activate pro-apoptotic proteins such as Bax, caspases-7, -8, and -9, while inhibiting the expression of cytoprotective proteins Bcl-2 and Bcl-x in MCF-7 cells [15]. However, the mechanism of eriocitrin based anti-angiogenesis has not been reported yet.

This study was designed to explore the anti-angiogenic effects of eriocitrin through in vitro and in vivo investigations, as well as to elucidate its underlying mechanism. Notably, this investigation represents a pioneering effort to characterize the anti-angiogenic properties of eriocitrin. Human umbilical vein endothelial cells (HUVECs) were cultured to establish an in vitro model for tube formation, allowing for the assessment of early stages of capillary blood vessel development. Furthermore, the angiogenic potential of eriocitrin was investigated in vivo through the utilization of the chick embryo chorionic membrane (CAM) assay. To clarify the mechanism involved in the anti-angiogenic impact of eriocitrin, we also determined whether eriocitrin could inhibit VEGFR2 and analyzed alterations in proliferation and apoptotic pathways.

## 2. Materials and Methods

### 2.1. Materials

Eriocitrin (≥98.0%, HPLC), dimethyl sulfoxide, gelatin, glutamine, acridine orange, ethidium bromide, and medium 199 were purchased from Sigma-Aldrich (St. Louis, MO, USA). Phosphate-buffered saline (PBS), trypsin-EDTA, and fetal bovine serum (FBS) were acquired from Moregate (Brisbane, Australia). MCDB-104 was purchased from Wako Pure Chemicals Industries (Osaka, Japan). Atelocollagen bovine dermis (type I collagen) was purchased from Koken (Tokyo, Japan). Epidermal growth factor (EGF) was purchased from BD Bioscience (Bedford, MA, USA) and human basic fibroblast growth factor (bFGF) (recombination) was purchased from Australia Biologicals (San Ramon, CA, USA). Phospho- or total forms of c-Raf, MEK 1/2, ERK 1/2, VEGFR2, PI3K, AKT, mTOR, caspase-9, caspase-3, PARP, and β-actin were obtained from Cell Signaling Technology (Danvers, MA, USA). MMP-2 and MMP-9 were purchased from Santa Cruz Biotechnology, Inc. (Santa Cruz, CA, USA). Lipid emulsion and fertilized chicken eggs were acquired from Dongkook Pharmacy Co. in Seoul, Korea and Pulmuone Farm in Bonghwa, Korea, respectively. Other chemicals were all bought from Sigma-Aldrich (St. Louis, MO, USA).

### 2.2. Cell Culture

Endothelial cells utilized in this study were generously provided by Prof. Toshiro Ohta from the Laboratory of Cell and Molecular Biology of Aging, Department of Food and Nutritional Sciences, University of Shizuoka (Shizuoka, Japan). HUVECs were cultured in growth medium (MCDB-104 medium supplemented with 10% FBS, 10 ng/mL epidermal growth factor, 100 g/mL heparin, 100 ng/mL endothelial cell growth factor, and 1% penicillin/streptomycin mixture) as previously reported [16] and incubated at 37 °C in 5% CO_2_ and 95% air.

### 2.3. Cell Viability Assay

The cytotoxic effects of eriocitrin to HUVEC cells were evaluated using a Cell Counting Kit-8 (Dojindo, Kumamoto, Japan). Briefly, HUVECs were planted into 96-well plates at a density of 1.0 × 10^4^ cells/well. After 24 h of incubation, eriocitrin was added to the plate at various concentrations (0, 25, 50, 100 μM) followed by incubation for 24 h. After CCK-8 solution was added (10 μL per well), cells were incubated for 4 h. Absorbance was then measured at 540 nm using a microplate reader.

### 2.4. Lactate Dehydrogenase (LDH) Release Assay

Cytotoxicity of eriocitrin was also evaluated using a Cytotoxicity LDH Assay Kit (Dojindo Laboratories, Kumamoto, Japan). HUVECs, at a density of 1.0 × 10^4^ cells/well in a 96-well plate, were treated with various concentrations of eriocitrin (0, 25, 50, 100 μM) for a duration of 48 h. A lysis buffer was subsequently introduced as a control for proliferation and allowed to incubate for 30 min. Then, the supernatant was relocated to a new 96-well plate, after which 100 μL of working solution was added. After incubating for 30 min, the stop solution was applied, and absorbance was recorded at 490 nm.

### 2.5. Tube Formation Assay

As previously described, following the induction of capillary-like structures in collagen gel, the effect of eriocitrin was evaluated using a tube formation assay [16]. Matrigel was divided into a 24-well plate (200 μL/well) and subsequently allowed to polymerize at 37 °C. Matrigel pre-coated wells were seeded with HUVECs (1.2 × 10^5^ cells/well) and incubated at 37 °C for 40 min. Following the removal of culture media from the wells, Matrigel was reapplied to each well and subjected to incubation at 37 °C for 30 min. HUVECs were subsequently exposed to eriocitrin (0, 25, 50, 100 μM) or just MCDB-104 media (control) for 24 h. This medium was supplemented with 0.5% FBS, 8 nM phorbol 12-myristate 13-acetate, 10 ng/mL bFGF, and 25 μg/mL ascorbic acid. The morphology of the tubes was documented using a phase-contrast microscope (Leica DMi8, Wetzlar, Germany) and the tube formation was quantified using the ImageJ program (version 1.8.0) to count tubes in each image (Media Cybernetics, Rockville, MD, USA).

### 2.6. Migration Assay

To evaluate the influence of eriocitrin on HUVEC cell migration, a wound healing assay was conducted using a published method [17]. HUVECs were cultured in 24-well plates at a cellular density of 1.4 × 10^5^ cells/well. The cellular monolayer was deliberately wounded using a plastic pipette tip and subsequently rinsed twice with PBS to eliminate dislodged cells. Subsequent to this preparation, the cells were exposed to treatments with eriocitrin at concentrations of 0, 25, 50, and 100 μM. Images capturing the process of wound healing 24 h post-treatment were acquired utilizing a phase-contrast microscope (Leica DMi8, Germany) under a magnification of ×100. Cell migration was quantified by determining the proportion of cells that transitioned from the initial wound perimeter towards the center (% of control).

### 2.7. Apoptosis

Apoptotic cell distribution was determined utilizing an Annexin V and Dead Cell Kit (Merck, Millipore, Burlington, MA, USA), following the guidelines provided by the manufacturer. Briefly, HUVECs underwent treatment with varying concentrations of eriocitrin (0, 25, 50, 100 μM), were subsequently harvested, and then resuspended in PBS to achieve a cell density of 2.3 × 10^5^ cells/mL. A single cell suspension was merged with 100 μL of Annexin V/Dead Cell reagent in separate aliquots, followed by incubation at ambient temperature in darkness for 20 min. Subsequent to an hour of treatment, H_2_O_2_ (50 μM) was utilized as a benchmark for comparative activity assessment. Subpopulations of apoptotic cells were identified utilizing a Muse^TM^ Cell Analyzer (Merck, Millipore, USA).

### 2.8. AO/EB Staining

Acridine orange/ethidium bromide (AO/EB) dual labeling was performed to determine the morphology of apoptotic cells. In short, HUVECs were cultured in a 24-well plate (3.0 × 10^4^ cells/well) and subsequently incubated at 37 °C. After the plate was washed carefully with PBS, cells were exposed to varying concentrations of eriocitrin (0, 25, 50, 100 μM) for 24 h. Each well was supplemented with a staining solution containing 50 μg/mL AO and 50 μg/mL EB. Subsequently, HUVECs were observed under a fluorescence microscope (Leica DMi8, Germany) with a magnification of 200×. Control groups included untreated cells as negatives and cells treated with 50 μM H_2_O_2_ as positives. A count of at least 200 cells was performed, with cells being classified into three morphological categories (normal cells, necrotic cells, and apoptotic cells).

### 2.9. TUNEL Assay

HUVECs exhibiting DNA fragmentation were detected in situ through the application of the terminal deoxynucleotidyl transferase-dUTP nick end labeling (TUNEL) assay. Apoptosis in human umbilical vein endothelial cells (HUVECs) was assessed using an In-situ Cell Death Detection Kit (Roche, Mannheim, Germany) in accordance with the manufacturer’s instructions. Briefly, HUVECs were cultured in a 24-well plate at a density of 3.0 × 10^4^ cells per well. Subsequent to a 24 h incubation at 37 °C to ensure adequate cell adhesion, the cells were exposed to differential concentrations of eriocitrin (0, 25, 50, 100 μM) over a 48 h. Subsequently, cells were fixed using 4% paraformaldehyde at 4 °C for 1 h. The plate underwent two PBS washes, followed by the addition of 100 µL of terminal deoxynucleotidyl transferase (TdT) reaction reagent to each well. Upon completion of the reaction process of the solution at a temperature of 37 °C for a duration of one hour, each well underwent a triple washing procedure using PBS. Subsequent to the application of 4’,6-diamidino-2-phenylindole (DAPI), and with a subsequent incubation period at ambient temperature for 1 h, cellular specimens were analyzed through the utilization of a fluorescence microscope (Leica Dmi8, Germany).

### 2.10. Chick Embryo Chorioallantoic Membrane (CAM) Assay

The CAM assay was executed in accordance with previously described methods [18]. Concisely, embryonic eggs were subjected to incubation within an incubator for a duration of four days at a temperature of 37.5 °C. To delineate the yolk sac from the exterior membrane, an aperture was fashioned at the constricted extremity of the egg, and 4 mL of albumin was aspirated from the egg utilizing a 20-gauge hypodermic needle. Eggs were put back in an incubator after the opening was resealed with a tape. On the next day, a volume of 10 μL of retinoic acid (5 nmol/egg), serving as a positive control, or eriocitrin (5–50 nmol/egg) was amalgamated with methylcellulose and subsequently administered onto silicon rings situated atop the CAM surface. Subsequent to a 2-day incubation period, the requisite quantity of a 20% lipid emulsion was administered into the CAM for the visualization of blood vessels. A minimum of 5 eggs were allocated for each experimental sample. This experiment was conducted in quintuplicate. The inhibition ratio (% of control) pertaining to the formation of new blood vessels within the area demarcated by a white ring, was subsequently determined.

### 2.11. Western Blot Analysis

HUVECs, quantified at 2.4 × 10^4^ cells per well, were embedded within a three-dimensional collagen matrix. Subsequently, these HUVECs underwent treatment with eriocitrin (25–100 μM) for periods of either 12 h or 24 h. Preparation of cell lysates was conducted in accordance with previously established protocols [19]. Proteins derived from tubule-forming HUVECs were subsequently fractionated using 8–10% sodium dodecyl sulfate-polyacrylamide gel electrophoresis (SDS-PAGE) and thereafter transferred onto polyvinylidene difluoride membranes. Protein bands demonstrating immunoreactivity were subjected to detection via an enhanced chemiluminescence (ECL) select Western blotting detection reagent, in strict adherence to the guidelines provided by the manufacturer. The ImageJ software application was utilized to quantitatively measure and graphically represent the intensity of proteins (NIH, Bethesda, MD, USA).

### 2.12. Statistical Analysis

Results were expressed as the mean ± standard error (SE) of three or five independent experiments, and were analyzed using one-way analysis of variance, followed by the Holm–Sidak method or Student’s *t*-test. Statistical significance was indicated by asterisk (* *p* < 0.05; ** *p* < 0.01; *** *p* < 0.001).

## 3. Results

### 3.1. Effect of Eriocitrin on Proliferation

The structure of eriocitrin is shown in Figure 1A. To assess its impact on HUVECs, we initially examined its influence on cell growth. HUVEC cells were incubated with either medium alone (control) or eriocitrin at various concentrations (25–100 µM) for 24 h (Figure 1B). Results demonstrated that eriocitrin did not exert any discernible effect on cell viability. Furthermore, 48 h treatment of eriocitrin at concentrations ranging from 25 to 100 µM did not induce significant LDH release into the supernatant (Figure 1C). These findings suggest that eriocitrin could reduce cell viability without eliciting cytotoxicity in HUVECs at the tested concentrations (25–100 µM), even under prolonged incubation time periods.

### 3.2. Effect of Eriocitrin on In Vitro Angiogenesis

In vitro angiogenesis experiments are a valuable method for investigating the effects of pharmaceutical agents on angiogenesis. To determine eriocitrin angiogenic potential, a tube formation assay was conducted using eriocitrin (25–100 μM). Following a 24 h incubation period, HUVECs cultured in the medium alone (control) exhibited a characteristic tube-like mesh organization, representing a typical arrangement indicative of tubulogenesis [20]. Figure 2A demonstrates that eriocitrin significantly inhibited the formation of tubular structures in a concentration-dependent fashion. Exposure to eriocitrin concentrations ranging from 25 to 100 µM significantly suppressed the tube formation of HUVECs stimulated by bFGF (Figure 2B). At a concentration of 25 μM, eriocitrin exhibited a slight reduction in tube width and inhibited the development of some branches. Small branches remained when eriocitrin was used for treatment at concentrations >25 μM. However, many cells underwent apoptosis and tubes started to separate. At concentrations of 0, 25, 50, and 100 μM, the ratios of tube area per image field were 64.6%, 53.3%, 43.4%, and 25.7%, respectively.

### 3.3. Effect of Eriocitrin on Migration

HUVEC cells were exposed to eriocitrin at different concentrations (25–100 µM) for a duration of 24 h. Eriocitrin treatment was found to inhibit the migration of HUVEC cells (Figure 3A). Following treatment with eriocitrin at concentrations of 25, 50, and 100 µM, the number of migrating cells was significantly reduced to 94.9%, 73.3%, and 54.2% (% of control), respectively (Figure 3B). Endothelial cell migration was significantly reduced, if not completely halted, in the high concentration group. These findings demonstrated that eriocitrin could considerably slow down migration of HUVEC cells depending on dose concentrations.

### 3.4. Analysis of Apoptosis by Annexin V/PI Staining

Flow cytometry was utilized to investigate the impact of eriocitrin on apoptosis of HUVEC cells following Annexin V and PI labeling. The annexin V/PI staining experiment involved two groups: the negative control group treated with medium, and the positive control group treated with H_2_O_2_. The apoptotic rates following treatment with eriocitrin (25–100 μM) or H_2_O_2_ (50 μM) are depicted in Figure 4A. Significantly different percentages of late apoptotic cells were observed between the positive control (20.3%) and negative control (0.9%). Following exposure to eriocitrin at increasing concentrations of 25, 50, and 100 μM, the percentages of late apoptotic cells were 7.5%, 13.2%, and 14.9%, respectively (Figure 4B). These findings indicate that eriocitrin can cause HUVEC cells to undergo evident apoptosis and that normal cells are progressively decreased in number when eriocitrin concentration is increased.

### 3.5. Analysis of Apoptosis by AO/EB Staining

HUVEC cells were dual-stained with AO/EB for 24 h after eriocitrin was added. As a result, in the negative control group without treatment with eriocitrin, all cells exhibited green nuclei without significant cell apoptosis (Figure 5A). However, in the positive control group treated with H_2_O_2_, necrotic cells (red cells) were generally observed. As the concentration of eriocitrin was increased, a decrease in the count of green-yellow nuclei, indicative of early apoptosis, was observed, whereas the count of orange nuclei, signifying late apoptosis, exhibited a gradual increase. Any red nuclei, meaning necrotic cells, were not observed with any concentrations of eriocitrin. The apoptotic cell percentage in the negative control group (5.5%) was significantly lower than that observed in the positive control (89.5%) (Figure 5B). In the presence of eriocitrin at concentrations of 25, 50, and 100 µM, percentages of apoptosis cells were 11.5%, 20.7%, and 32.5%, respectively.

### 3.6. Analysis of Apoptosis by TUNEL Staining

To examine the apoptotic effect of eriocitrin, TUNEL labeling was performed. TUNEL-positive HUVECs were detected after treatment with eriocitrin for 24 h (Figure 6A). Eriocitrin caused DNA cleavage in a dose-dependent manner (25–100 μM). In comparison to the negative control (14.0%), there was a significant increase in the percentage of apoptotic HUVECs in the positive control (97.6%). In the presence of eriocitrin at concentrations of 25, 50, and 100 µM, percentages of apoptosis cells were 19.8%, 30.9%, and 53.6%, respectively (Figure 6B).

### 3.7. Inhibitory Effect of Eriocitrin on Angiogenesis in CAM

In order to elucidate the influence of eriocitrin on angiogenesis in vivo, a chick embryo chorionic membrane (CAM) assay was performed. CAM results revealed that eriocitrin suppressed neovascularization in chick embryos, thereby inhibiting embryonic angiogenesis (Figure 7A). Eriocitrin demonstrated a considerably greater inhibitory activity on CAM microvessel production in comparison with the control group. Eriocitrin at increasing concentrations (5, 10, and 50 nmol/egg) dose-dependently inhibited chorioallantoic vessel formation (by 31.4%, 55.7%, and 74.7%, respectively) compared to the negative control (Figure 7B). At a concentration of 50 nmol/egg, eriocitrin significantly blocked the angiogenic process in the CAM. In particular, eriocitrin at a concentration of 50 nmol/egg demonstrated anti-angiogenic effects comparable to those of retinoic acid. These results indicate that eriocitrin exhibits promising anti-angiogenic activities both in vitro and in vivo.

### 3.8. Effects of Eriocitrin on Raf/MEK/ERK Pathways

To determine how eriocitrin could control angiogenesis and shed further light on the process by which eriocitrin inhibited the proliferation of HUVEC cells, molecular analysis was conducted (Figure 8A). Results showed that eriocitrin treatment for 12 h significantly reduced expression levels of p-c-Raf/c-Raf, p-MEK 1/2/MEK 1/2, and p-ERK1/2/ERK 1/2 compared to the control (Figure 8B). In particular, eriocitrin at a concentration of 50 µM or more significantly reduced their expression levels. Therefore, the suppression of angiogenesis by eriocitrin may primarily involve the inhibition of the Raf/MEK/ERK signaling pathway, thereby inhibiting the migration and proliferation of HUVECs.

### 3.9. Effects of Eriocitrin on VEGFR2-Mediated Downstream Signaling Pathways

To further explain the anti-angiogenic effect of eriocitrin, expression levels of VEGFR2 and its downstream pathway PI3K/AKT/mTOR after eriocitrin treatment were determined. Western blot analysis was performed to quantify the levels of p-VEGFR2 and total VEGFR2 in HUVEC cells following eriocitrin treatment (Figure 9A). The results indicated that eriocitrin dose-dependently suppressed VEGFR2 tyrosine phosphorylation (Figure 9B). However, total level of VEGFR2 expression showed minimal variation. In addition, it was found that phosphorylation of PI3K/AKT/mTOR, a sub-signal, was significantly inhibited after inhibiting p-VEGFR2 by eriocitrin (25–100 μM). Therefore, eriocitrin can inhibit the survival of HUVECs, thereby inhibiting angiogenesis.

### 3.10. Effects of Eriocitrin on Caspase Pathways

Western blotting was utilized to evaluate the expression levels of apoptosis-related proteins in HUVECs and assess whether or not cells were undergoing apoptosis in response to treatment with eriocitrin at different concentrations for 24 h (Figure 10A). Results showed that treatment with eriocitrin activated caspase-9, -3, and PARP and increased protein levels of their activated types (cleaved caspase-9, -3, and PARP) depending on dose concentrations (Figure 10B). Eriocitrin also suppressed angiogenesis by triggering apoptosis primarily via protease that could cleave caspase-9, caspase-3, and PARP.

### 3.11. Effects of Eriocitrin on MMP-2 and MMP-9 Protein Expression

It is recognized that extracellular proteases, particularly matrix metalloproteinases (MMPs), play a pivotal role in tumor movement and penetration by promoting angiogenesis. Thus, expression of MMPs (MMP-2 and MMP-9) were examined following treatment with eriocitrin at various concentrations for 24 h. As shown in Figure 11, expression levels of MMP-2 and MMP-9 declined as the concentration of eriocitrin increased (25–100 μM), further suppressing vascular endothelial cells’ activity and impairing their motility.

## 4. Discussion

Angiogenesis is one of the key conditions for proliferative infiltration and metastasis of cancer. Anti-angiogenic therapy is still an anti-tumor strategy. VEGFR2 undergoes autophosphorylation and signaling, which strongly activates common downstream signaling pathways such as PI3K/AKT/mTOR and Raf/MEK/ERK. VEGFR2 is a popular therapeutic target. It occupies the majority of studies on angiogenesis inhibitors. However, some common problems such as treatment efficacy, reproducibility, and popularization of treatment methods cannot be ignored. It remains challenging to solve these problems. To overcome them, we focused on eriocitrin, a Hanseong branch of flavonoid flavanone [21].

Flavonoids and flavonoid derivatives have been suggested as the most promising anti-angiogenic substances [22]. The discovery of less toxic and cost-effective molecules is very important in cancer angiogenesis management [22]. In particular, anti-angiogenic effects of flavanones have been proven by a number of studies [23,24]. Therefore, we systematically investigated the anti-angiogenic effect of eriocitrin, a flavanones component of lemon and peppermint. Such effect has not been reported so far.

There are reports indicating that eriocitrin is capable of effectively inhibiting the proliferation of cancer cells [25]. In addition, eriocitrin has no toxicity to vascular endothelial cells. Our experimental results have shown that in vitro angiogenesis can be significantly inhibited by non-toxic concentrations of eriocitrin. Previous studies have shown that eriocitrin can inhibit cancer movement by inhibiting cell movement during the wound healing process in A549 and H1299 cancer cells [9]. Our results also suggest that eriocitrin can help prevent tumor growth by inhibiting capillary formation and migration during the initial phases of angiogenesis. Wang et al. have shown apoptosis effects of eriocitrin in HepG2 cells and Huh7 cells [25]. Therefore, we determined possible mechanisms involved in eriocitrin-induced death of HUVECs.

These results suggest that the inhibition of tube formation and proliferation of HUVECs caused by eriocitrin is associated with the induction of apoptosis. In addition, eriocitrin significantly reduced neovascularization of the chick embryonic CAM model without showing toxicity to existing blood vessels. These findings lead us to conclude that eriocitrin possesses anti-angiogenic activity both in vitro and in vivo.

In this study, the mechanism involved in the anti-angiogenesis effect of eriocitrin was examined. At the molecular level, the process by which eriocitrin regulated angiogenesis in HUVECs was studied. Inhibition of MAPK-ERK in the tumor’s vascular system can inhibit angiogenesis, reduce tumor growth, and allow for MAPK-ERK signaling, endothelial cell spread, and survival during angiogenesis [26,27]. In addition, eriocitrin could inhibit tumor growth by suppressing movement and survival of vascular endothelial cells through inhibition of MAPK-ERK phosphorylation. The results of this study indicate that eriocitrin exhibits a dose-dependent inhibition of VEGFR2 phosphorylation in HUVECs, suggesting its anti-angiogenic effect on VEGFR2 as a target. Then, the expression levels of angiogenesis-related proteins, serving as downstream effectors of the VEGFR2 signaling pathway, were quantified. During angiogenesis, PI3K/AKT/mTOR is activated in endothelial cells and these phosphorylated PI3K/AKT/mTOR generate angiogenesis factor in tumor cells, which is involved in the cell proliferation and survival [28,29]. Studies have indicated that PI3K/AKT deactivation via eriocitrin can inhibit tumor initiation, progression, and angiogenesis by inhibiting phosphorylation of various downstream substrates such as mTOR.

In HUVECs, expression levels of caspase-9, -3, and PARP proteins linked to apoptosis were evaluated. Proteolytic cleavage can induce significant alterations in cell morphology, including membrane blebbing, DNA fragmentation, exposure of phosphatidylserine on the cell surface, and the generation of apoptotic vesicles [30]. Figure 10 shows that treatment with eriocitrin activated caspase-9, -3, and PARP and elevated protein levels of activated caspase-9, -3, and PARP. Thus, eriocitrin can cause HUVEC cells to undergo apoptosis, thus preventing angiogenesis. It is believed that different substrate and basilar membrane components must break down for malignant neoplasms to proliferate.

MMPs degrade numerous components of the extracellular matrix. Gelatinous substrates and type IV collagen can be broken down by MMP-2 and MMP-9 [31,32]. Eriocitrin prevented vascular endothelial cells from expressing or using MMP-2 and MMP-9. By inhibiting the production of extracellular collagen and the breakdown of gelatinous substrates, eriocitrin can lower angiogenesis and cell migration.

In summary, this study comprehensively elucidated the anti-angiogenic effects and underlying mechanisms of eriocitrin using HUVEC cells. Eriocitrin demonstrated potent therapeutic efficacy and inhibited angiogenesis in HUVEC cells by modulating the MAPK/ERK, VEGF2, PI3K/AKT/mTOR, and caspase pathways. This study supports further development of eriocitrin as a novel anti-angiogenic drug.

## 5. Conclusions

This study confirms that eriocitrin has an anti-angiogenesis effect in HUVEC cells. Eriocitrin induced apoptotic pathways in endothelial cells, simultaneously suppressing HUVEC tube formation and migration. Eriocitrin effectively suppressed neovascularization within the chorioallantoic membrane of developing chick embryos through in vivo angiogenesis. Additionally, eriocitrin inhibited phosphorylation activity of the MAPK/ERK pathway. Eriocitrin significantly inhibited VEGFR2, which is recognized as playing a crucial role in the process of angiogenesis. It also mediated downstream signaling pathway PI3K/AKT/mTOR to inhibit angiogenesis. Furthermore, results of inhibiting the activity of vascular endothelial cells were derived through the inhibition of MMP-2 and MMP-9.

In conclusion, eriocitrin exhibited anti-angiogenic effects through by inhibiting tube formation, migration, survival, and apoptosis. This study also demonstrated relevant molecular mechanisms involved in its effects. We also discovered that eriocitrin, a kind of flavanone, might play a role in the treatment of angiogenesis-mediated illnesses as a food and/or natural medication.

## Figures and Tables

**Figure 1 nutrients-16-01091-f001:**
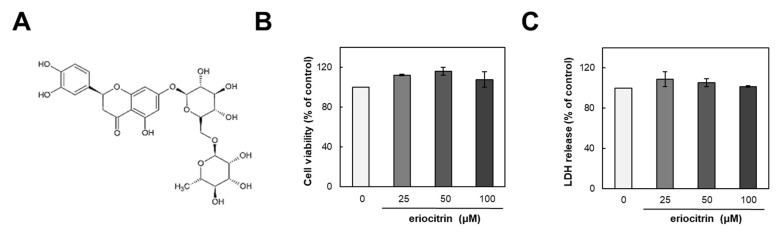
Chemical structure of eriocitrin and its effects on proliferation of HUVECs. (**A**) Chemical structure of eriocitrin. After cells were cultured with eriocitrin at different concentrations (25–100 µM), (**B**) cell viability and (**C**) cytotoxicity were assessed. Data are shown as mean ± SE, n = 3.

**Figure 2 nutrients-16-01091-f002:**
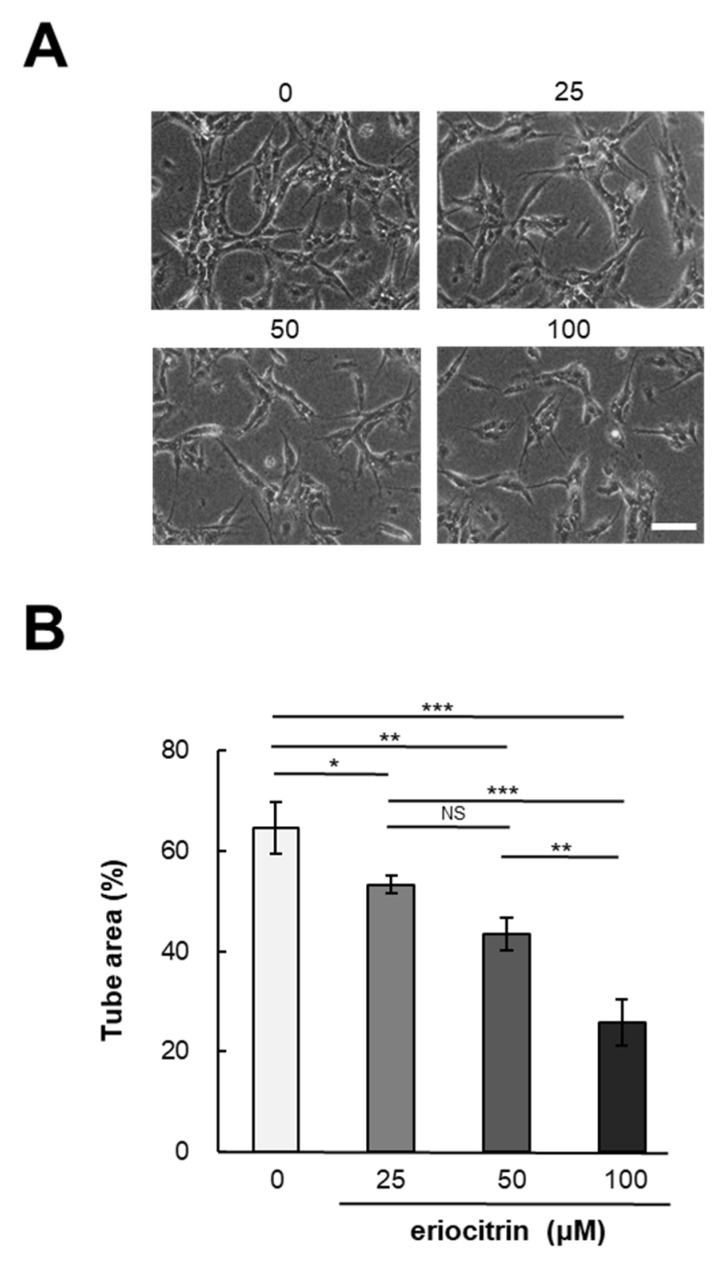
Eriocitrin attributes the formation of tubular structures in HUVECs. (**A**) HUVECs were cultured within a sandwich configuration of collagen gels, exposed to various concentrations of eriocitrin (25–100 µM). The bar indicates 100 μm. (**B**) The extent of tubular structure formation was quantitatively assessed. Data are shown as mean ± SE, n = 3. * *p* < 0.05; ** *p* < 0.01; *** *p* < 0.001; NS, non-significant vs. among groups.

**Figure 3 nutrients-16-01091-f003:**
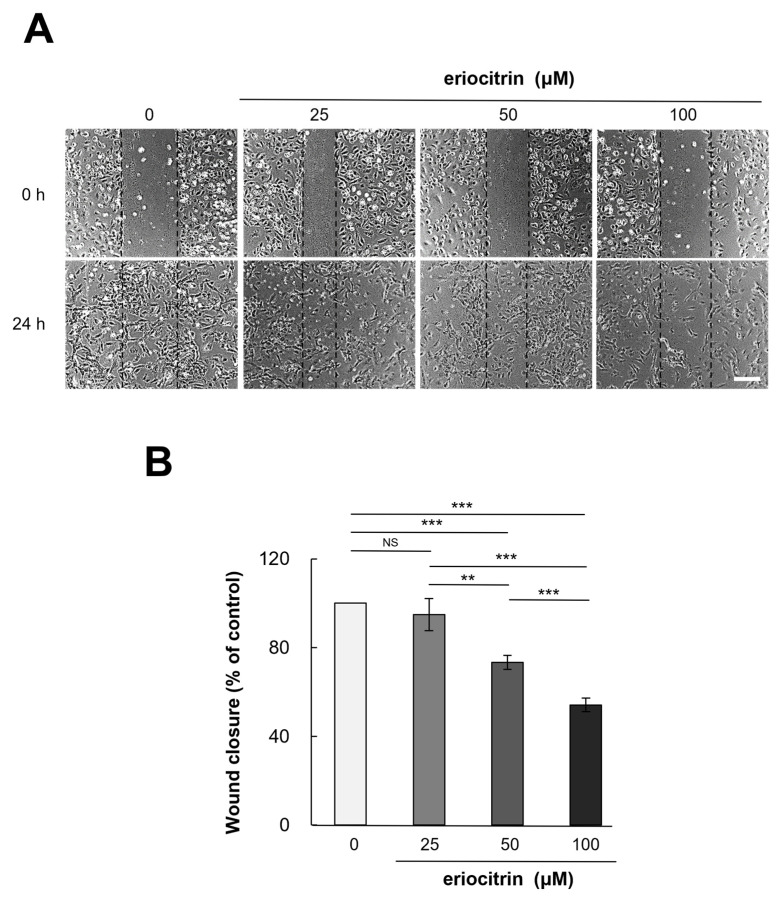
Eriocitrin suppresses the migration of HUVECs in vitro. (**A**) A monolayer of HUVECs was mechanically disrupted to create a linear wound, followed by exposure to differing concentrations (25–100 µM) of eriocitrin for a duration of 24 h. The bar indicates 200 μm. (**B**) The extent of HUVEC migration into the wounded area at the 24 h mark was quantified. Data are shown as mean ± SE, n = 3. ** *p* < 0.01; *** *p* < 0.001; NS, non-significant vs. among groups.

**Figure 4 nutrients-16-01091-f004:**
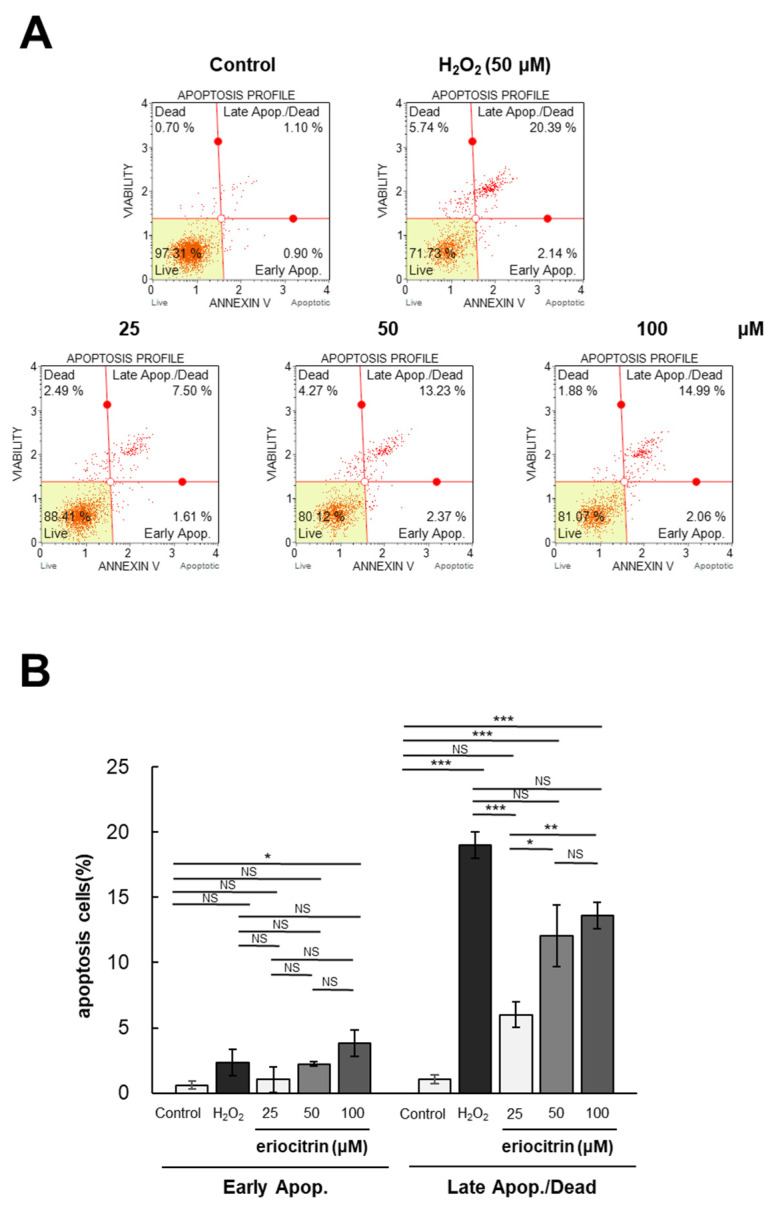
Eriocitrin triggers apoptotic cell death in HUVECs. (**A**) The induction of apoptosis in HUVECs by eriocitrin over a 24 h period was assessed using flow cytometry subsequent to staining with annexin V-fluorescein isothiocyanate (FITC) and PI. (**B**) The percentage of apoptotic cells relative to the total cell population is quantified. Data are shown as mean ± SE, n = 3. * *p* < 0.05; ** *p* < 0.01; *** *p* < 0.001; NS, non-significant vs. among groups.

**Figure 5 nutrients-16-01091-f005:**
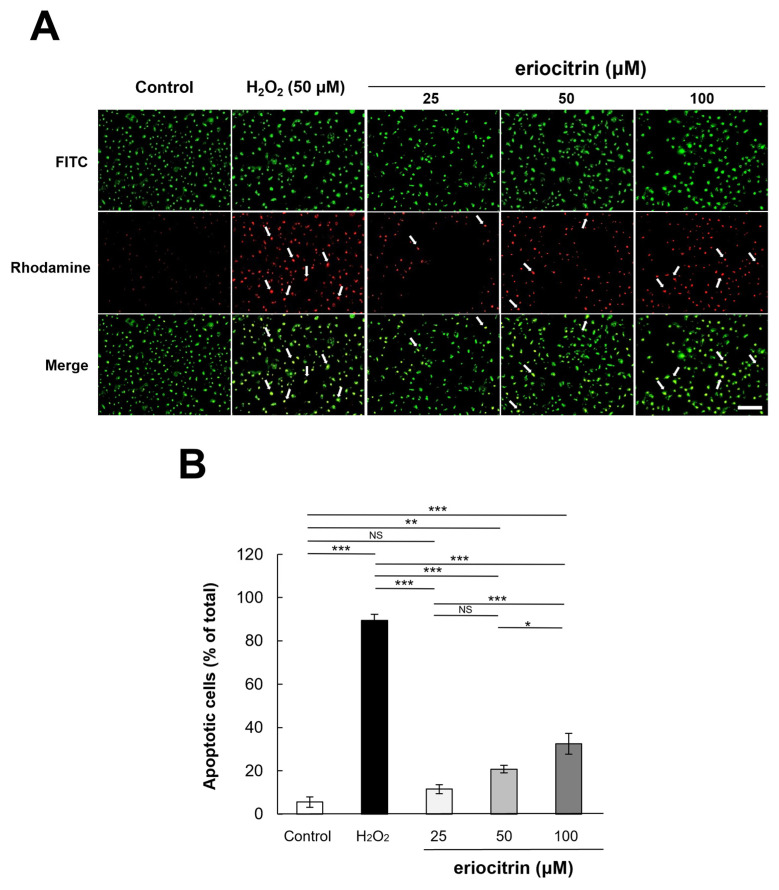
Analysis of apoptotic morphological changes through AO/EB staining. (**A**) Comparing fluorescently dyed HUVECs with varied concentrations of eriocitrin (25–100 μM) to untreated and positive (H_2_O_2_) controls are shown in representative photographs. Apoptosis in HUVECs exposed to eriocitrin for 24 h was assessed using AO/EB staining. The bar indicates 200 μm. (**B**) The percentage of apoptotic cells relative to the total cell population is quantified. Data are shown as mean ± SE, n = 3. * *p* < 0.05; ** *p* < 0.01; *** *p* < 0.001; NS, non-significant vs. among groups.

**Figure 6 nutrients-16-01091-f006:**
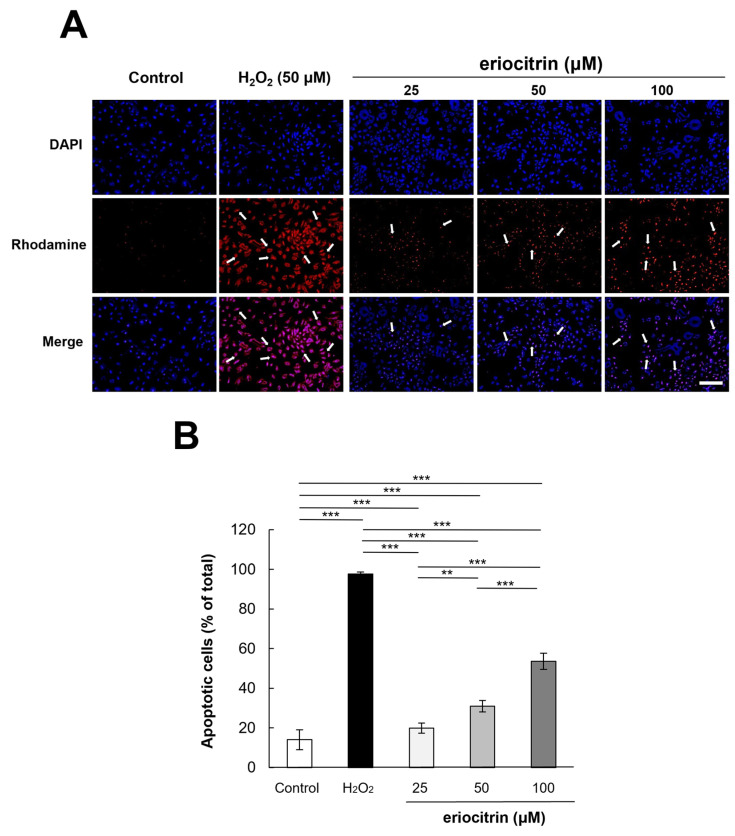
Effect of eriocitrin on apoptosis as evidenced by TUNEL staining. (**A**) Compared to untreated and positive (H_2_O_2_) controls, fluorescent microscopy images depict human umbilical vein endothelial cells (HUVECs) subjected to various concentrations of eriocitrin (25–100 μM). TUNEL labeling is used to measure apoptosis in HUVECs exposed to eriocitrin for 48 h. Rhodamine (red) and DAPI (blue) as markers for TUNEL-positive cells are used to identify their nuclei. The bar indicates 200 μm. (**B**) The percentage of apoptotic cells relative to the total cell population is quantified. Data are shown as mean ± SE, n = 3. ** *p* < 0.01; *** *p* < 0.001 vs. among groups.

**Figure 7 nutrients-16-01091-f007:**
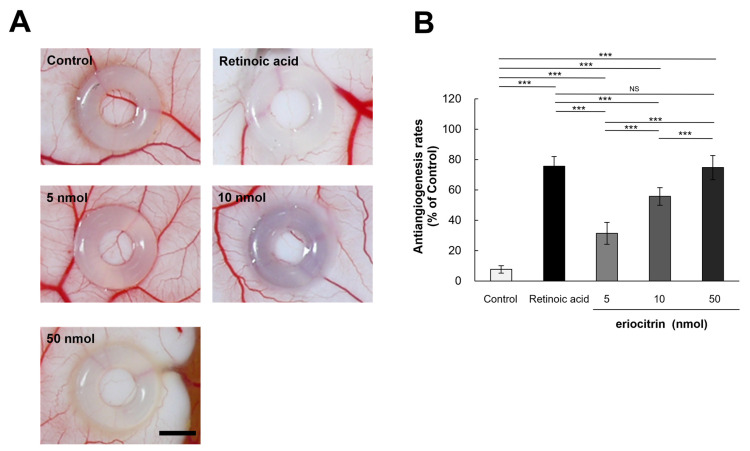
Eriocitrin attenuates embryonic angiogenesis as observed in the CAM assay. (**A**) Eriocitrin exhibits an inhibitory influence on tumor-induced angiogenesis within an in vivo experimental setup dedicated to studying angiogenesis. The bar indicates 2 mm. (**B**) Administration of retinoic acid (5 nmol/egg retinoic acid) or eriocitrin (ranging from 5 to 50 nmol/egg) resulted in a diminished rate of neovascular formation. The experiment was repeated at least five times and representative data are shown. *** *p* < 0.001; NS: non-significant vs. among groups.

**Figure 8 nutrients-16-01091-f008:**
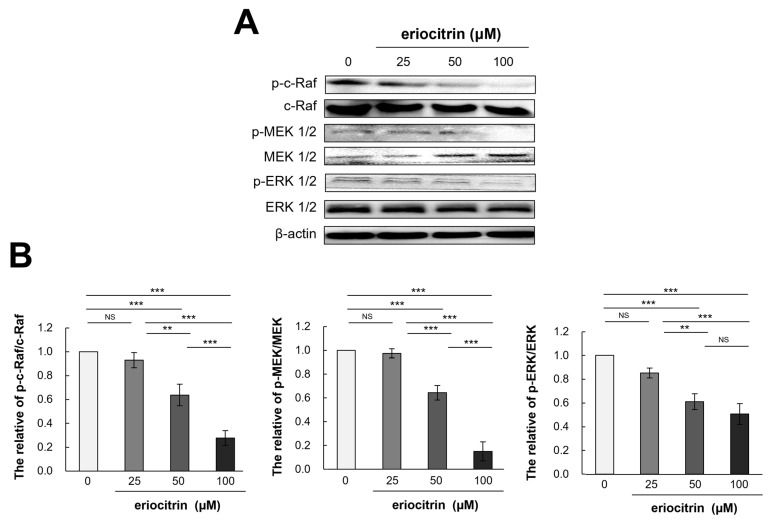
Eriocitrin deactivates the phosphorylation of c-Raf, MEK1/2, and ERK 1/2. (**A**) Proteins were harvested from tube-forming HUVECs subjected to treatment with various concentrations of eriocitrin (25–100 μM) for 12 h. Alterations in the expression levels of of c-Raf, MEK 1/2, and ERK 1/2 were examined via Western blot analysis. (**B**) The experiment was repeated at least three times and representative data are shown. ** *p* < 0.01; *** *p* < 0.001; NS, non-significant vs. among groups.

**Figure 9 nutrients-16-01091-f009:**
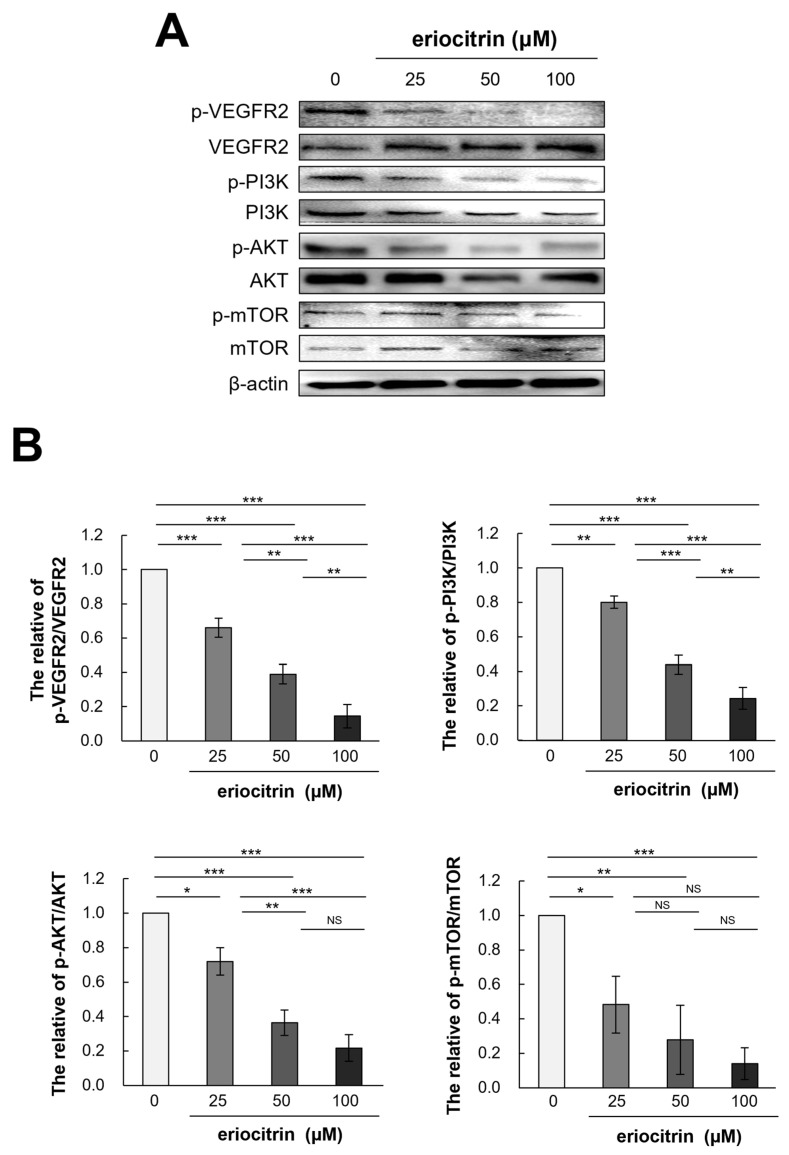
Eriocitrin impedes the phosphorylation of VEGFR2 and attenuates the activation of downstream signaling cascades mediated by VEGFR2. (**A**) Expression profiles of total and phosphorylated forms of proteins associated with the VEGFR2 and PI3K/AKT/mTOR signaling pathways in HUVECs, following exposure to various concentrations of eriocitrin (25–100 μM) for 24 h, were evaluated via Western blot analysis. (**B**) The experiment was repeated at least three times and representative data are shown. * *p* < 0.05; ** *p* < 0.01; *** *p* < 0.001; NS, non-significant vs. among groups.

**Figure 10 nutrients-16-01091-f010:**
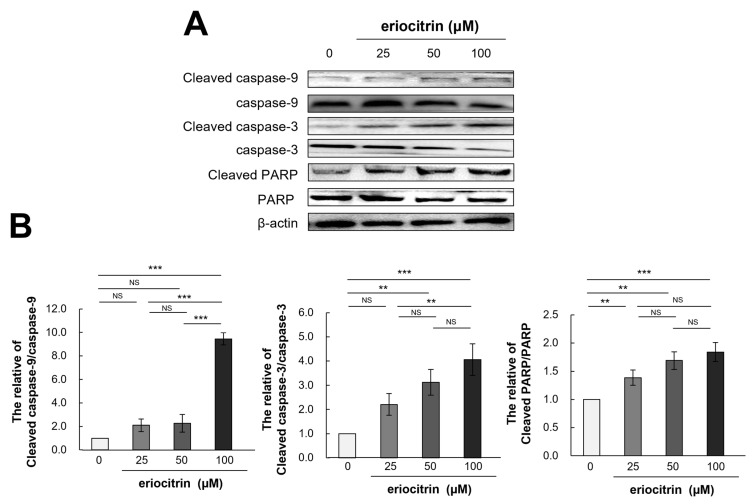
Eriocitrin promotes apoptosis in tube-forming HUVECs by activating pro-apoptotic signals such as caspases. (**A**) Proteins were harvested from tube-forming HUVECs subjected to treatment with various concentrations of eriocitrin (25–100 μM) for 24 h. Alterations in the expression levels of caspase-9, caspase-3, and PARP were examined via Western blot analysis. (**B**) The experiment was repeated at least three times and representative data are shown. ** *p* < 0.01; *** *p* < 0.001; NS, non-significant vs. among groups.

**Figure 11 nutrients-16-01091-f011:**
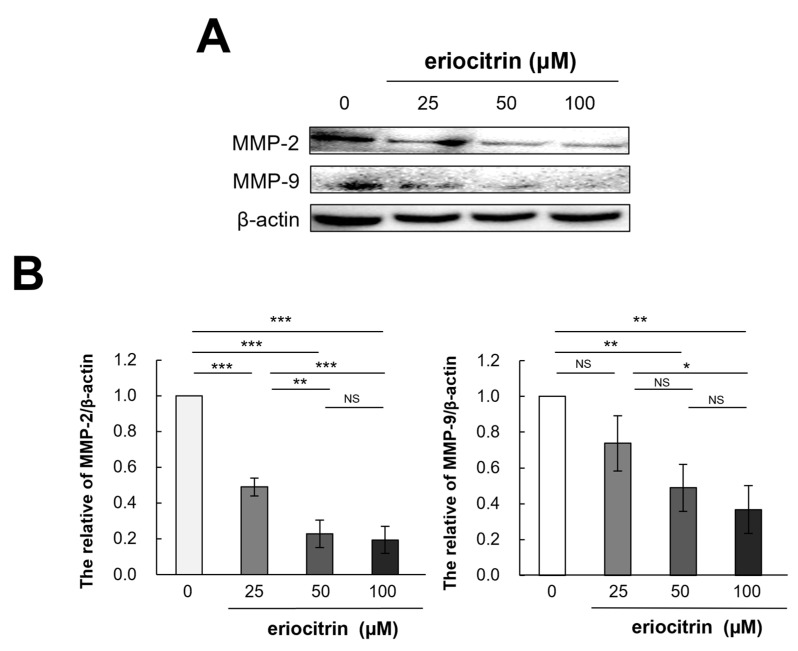
Eriocitrin reduces the expression of MMPs protein. (**A**) The expression levels of MMP-2 and MMP-9 proteins were quantified using Western blot analysis and normalized to β-actin level. (**B**) The experiment was repeated at least three times and representative data are shown. * *p* < 0.05; ** *p* < 0.01; *** *p* < 0.001; NS, non-significant vs. among groups.

## Data Availability

Data are contained within the article.

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
