# Peer review of "Eriocitrin Inhibits Angiogenesis by Targeting VEGFR2-Mediated PI3K/AKT/mTOR Signaling Pathways"

_nutrients, 2024, doi:10.3390/nu16071091_

Round 1

Reviewer 1 Report

Comments and Suggestions for Authors

Review of manuscript nutrients-2927205

Title: Eriocitrin inhibits angiogenesis by targeting VEGFR2-mediated PI3K/AKT/mTOR signaling pathways

Authors: Ji-Yoon Baek, Jeong-Eun Kwak, Mok-Ryeon Ahn

General comment: This study explores the anti-angiogenic effects of flavanone eriocitrin through in vitro and in vivo approaches, as well as attempts to elucidate its underlying molecular mechanism. The authors use human umbilical vein endothelial cells (HUVEC) to establish an in vitro model for tube formation/angiogenesis, for the evaluation of early stages of capillary blood vessel development. Furthermore, the antiangiogenic potential of eriocitrin was investigated in vivo through the utilization of the chick embryo chorionic membrane assay. Finally, the potential effects through inhibition of VEGFR2 and proliferation and apoptotic pathways are investigated. The authors conclude that eriocitrin effectively suppressed angiogenesis in cultered HUVEC cells and neovascularization within the chorioallantoic membrane of developing chick embryos. Further, the flavanone regulates downstream signaling pathway PI3K/AKT/mTOR and increases endothelial cell apoptosis to inhibit angiogenesis. Hypothesis and objectives are sound and clearly exposed, methods seem adequate and results are well presented and fairly discussed. The findings might have a potential translational application to humans. Major concerns are the incomplete/incorrect statistical analysis of data and some contradictory data regarding western blot pictures and densitometry data. Some specific comments are detailed below:

Specific comments:

1)      Lines 38-41; sentence needs to be rewritten.

2)      Line 82; origin (purchase/gift) or isolation and preparation of HUVEC cells must be stated.

3)      Figure 1; although the authors state in the legend that cells were cultured with eriocitrin at different concentrations (25–100 μM) for 24 h and 48 h, data for cell viability and LDH for a single time point are shown in the figure. The authors should identify the data or else show data for both time periods.

4)      Figure 2B, incomplete statistical analysis; data from different concentrations of eriocitrin should be compared among them, not only to controls, in order to ensure a dose-dependent response. The authors should not state that there is a concentration-dependent fashion until data from the three doses are statistically compared and significant differences show up. A different number of asterisks in the comparison to controls does not indicate a significant difference between two data. The same comment applies to the rest of figures.

5)      The authors utilize three different methods to evaluate apoptosis, Annexin V/PI staining, AO/EB staining and TUNEL staining, and the three with a very similar result. Since they later confirm activation of apoptosis at molecular level by caspases 3 and 9 and PARP, the use of three different biochemical assays to evaluate apoptosis seems unnecessary.

6)      Figure 7B; contrary to the rest of figures, control data are not shown in this plot. Besides, it is highly surprising that there is no significant effect by 5 nmol eriocitrin with a 30% antiangiogenic effect. Perhaps the data should be double checked, or else number of data seems insufficient.

7)      Figure 8; picture of western blot should be representative of the densitometry data. Band density of p-ERK 1/2 in blot (panel A) shows higher intensity at 25 and 50 uM eriocitrin than in controls, even considering loading controls, but this is not sustained in densitometry data depicted in panel B. The authors should find a blot more representative of the final data or else repeat densitometry calculation or the assay.

8)      Figure 10, same as in figure 8; lack of correspondence between band intensity in blots and densitometry data. This is especially noteworthy for cleaved PARP, which band intensity for 100 uM is much higher than that of 50 uM in panel A, but barely no difference in panel B.

9)      Figure 11, same as in figures 8 and 10; band intensity for MMP-2 is very similar for all three tested concentrations of eriocitrin, as well as for actin loading control (panel A), but there seems to be significant differences between 25 uM and the other two doses in panel B. Blots depicted should be genuinely representative of the densitometry of pooled data.

10)   Line 396; it should say flavanones.

11)   Lines 399-400; the authors quote reference 25 to support that eriocitrin is capable of effectively inhibiting the proliferation of cancer cells, but such reference deals with eriocitrin metabolism in vivo and in vitro based on an efficient UHPLC-Q-TOF-MS/MS strategy. However, reference 26 addresses proliferation of human hepatocellular carcinoma cells through inducing apoptosis and arresting cell cycle, which seems rather more appropriate. This reference should be double checked.

12)   Line 421; the authors should not state that eriocitrin exhibits a dose-dependent inhibition of VEGFR2 phosphorylation in HUVECs until a complete statistical comparison has been performed.

13)   Line 435, same as above; no dose-dependent activated caspase-9, -3, and PARP until data have been properly compared.

Comments on the Quality of English Language

No further comment

Author Response

We are thankful for the reviewers’ constructive comments that helped to considerably improve and clarify the manuscript. We hope that its revised version answers their concerns. In the following we illustrate how we took the reviewers’ comments into account. Each reviewer is addressed individually, with the reviewer’s comments in italic font, our answers in normal font. We also made changes to the manuscript that is independent of the reviewers’ comments. 

Reviewer 2 Report

Comments and Suggestions for Authors

                The authors report that eriocitrin, a flavanone found in peppermint and citrus fruits, exerts an antiangiogenic action; inhibits the  proliferation, tube formation, migration, and apoptosis in HUVECs; suppressed the formation of new blood vessels; and inhibited PI3K/AKT/mTOR signaling pathway.

The authors are to be praised for the noivelty of these findings as there is almost no information about eriocitrin. Besdies, the study is deeply mechanistic as the different pathophysiological mechanisms (Apoptosis, angiogenesis,…) are investigated. The results are solid, the discussion is balanced, the conclusions are supported by the data, and the manuscript reads well.is

                The authors are to be praised for

-          All results should be confirmed by two different techniques. Therefore, besides TUNEL data, please also provide additional apoptosis techniques such as BAx/Bcl-2 ratio or cleaved caspase-3.

-          Fig 3A and 4A are not of good quality. Pleawse provide more convincing images

-          Akt is also implicated in adverse cardiac remodeling (Circulation 2016 Mar 8;133(10):954-66). The authors should mention that the inhibitoriy effecy of eriocitrin on Akt could also ameliorate adverse cardiac remodeling.

Comments on the Quality of English Language

Minor Review

Author Response

(The authors gave the same response as above.)

Round 2

Reviewer 1 Report

Comments and Suggestions for Authors

The authors have conveniently addressed all my comments and queries; therefore, I recommend to accept the revised version of the manuscript for publication at Nutrients.